# Physical Layer Security Performance of Multi-User Mixed Radio-Frequency/Free-Space-Optics System Based on Optimal User Interference

**DOI:** 10.3390/s23146523

**Published:** 2023-07-19

**Authors:** Zihe Shen, Yi Wang, Jiamin Wu

**Affiliations:** 1Key Laboratory of Electromagnetic Wave Information Technology and Metrology of Zhejiang Province, College of Information Engineering, China Jiliang University, Hangzhou 310018, China; p21030854041@cjlu.edu.cn (Z.S.); p22030854075@cjlu.edu.cn (J.W.); 2State Key Laboratory of Coal Resources and Safe Mining, China University of Mining and Technology, Xuzhou 221116, China

**Keywords:** RF/FSO, physical layer security, Fischer–Snedecor F distribution, multi-user, secrecy outage probability, average secrecy capacity

## Abstract

This paper presents research on the physical layer security performance of a multi-user mixed RF/FSO system based on optimal user interference. In this system model, the RF link experiences Rayleigh fading, and the FSO link follows the Fischer–Snedecor F distribution. The system adopts a double-hop-decode-and-forward (DF) relay scheme. We also consider the effect of directivity errors in the FSO link and assume the presence of an illegal eavesdropper with a single antenna near the RF link. The source node controls the energy collection and information forwarding using a multi-user structure based on simultaneous wireless information and power transfer (SWIPT). We select the optimal user to jam the eavesdropper’s communication. We derive closed-form expressions for the mixed RF/FSO communication system’s secrecy outage probability (SOP) and average secrecy capacity (ASC). Monte Carlo simulations are performed to verify the accuracy of these expressions. By formulating and simulating the simulation system, the impact of various important factors on the mixed system’s physical layer security (PLS) is analyzed. The analysis indicates that increasing the number of antennas and interference signal-to-noise ratio (SNR) of the optimal user, the time allocation factor and energy conversion efficiency, and the improvement in the quality of atmospheric channels with improved weather will significantly enhance this system’s PLS.

## 1. Introduction

The rapid scientific and technological development, and the growing network demand are overwhelming the radio frequency spectrum resources. Compared with traditional radio frequency (RF) communication, FSO (Free Space Optics) is a well-established wireless optical transmission technology. FSO communication has attracted a lot of interest as a potential solution to the short-range 5G wireless communication problem [1,2]. FSO communication has the advantages of being license-free and highly secure, as well as a low deployment cost and high traffic rate transmission, which is often used to solve the “last-kilometer” problem [3,4]. FSO provides a wide range of applications in outdoor and indoor services, for example, wireless video surveillance, data centers, terrestrial transmission, LAN connectivity, mobile cellular networks, last-mile solutions, space communications, radio astronomy, remote sensing, etc. [5,6]. However, FSO links are easily affected by directional errors, weather conditions, atmospheric turbulence and other factors, resulting in reduced communication quality or even interruption of communication. Therefore, although it has strong directivity and high security, it is not suitable for long-distance communication [7,8].

RF and FSO communication have their own advantages and disadvantages. Therefore, there are many technologies proposed to solve these problems. Using both RF and FSO technology is a very effective solution. This combination includes two possible configurations. The first one is the hybrid RF/FSO, in which an RF link is incorporated in parallel with an FSO link. The second combination is the mixed RF/FSO communication, in which RF communication is used at one hop and FSO communication at the other in dual-hop or relay-assisted networks [9]. We choose the mixed RF/FSO communication system to solve the aforementioned problems. The combination of RF communication reliability and FSO communication security under bad weather conditions can lead to better safety communication and also extend the information transmission distance [10]. The RF communication can be an easy target of eavesdropping by illegal eavesdroppers, because the RF link in the mixed communication system is a broadcast link. This problem seriously affects the security performance of the communication system; therefore, we further consider the mixed system’s physical layer security (PLS) performance. Secure communication can be achieved by just using the physical characteristics of wireless channels because the PLS does not need to rely on an enciphering algorithm. The PLS also has a low rate of complexity. Therefore, the mixed RF/FSO system’s PLS has received a significant amount of attention [11,12].

Compared with a single user, multiple users are usually more suitable for practical communication scenarios. The multiple users can improve the system throughput fault tolerance and reliability of RF links. For example, El-Malek et al. studied a multi-user mixed RF/FSO link model with a single input and multiple outputs. In particular, the FSO link is based on the Gamma–Gamma distribution, and the maximum proportional combination and selective combination schemes are used in the multi-antenna repeater, where the system selects the worst user to interfere with the eavesdropper [13]. The authors also studied the influence of RF channel co-frequency interference on the security performance of multi-user mixed RF/FSO communication systems under eavesdropping attack conditions [14]. Hu et al. studied the optimization problem based on the secrecy rate in multi-user mixed RF/FSO spectrum sharing networks [15].

The aforementioned studies are the only ones that consider multiple users in the PLS of mixed RF/FSO communication systems. The optimal user transmission and worst user interference schemes are used in these studies. The optimal user transmission is actually a single user transmission scheme, which can be easily affected by the external electromagnetic wave interference and the co-channel interference. The worst user cooperative jamming scheme has certain limitations, because it is limited by the transmitting power of the worst user and, therefore, the interference SNR is low and the interference effect on eavesdroppers is not obvious.

The Gamma–Gamma distribution is used for the FSO links in the above security performance studies of the mixed RF/FSO communication system with multi-user diversity. This distribution is mainly applicable to moderate and strong turbulence in the FSO link. Recently, an increasing amount of attention has been paid to the F distribution [16]. Peppas et al. proposed to use the F distribution to simulate the atmospheric turbulence on the FSO links. The results showed that the F distribution was more suitable for FSO fading models under all turbulence conditions, and the simulated experimental data matched closely with those in the actual communication scenario [17]. 

The F distribution is used to simulate the FSO link fading. Compared with the Gamma–Gamma and M distributions, it is more suitable for all turbulence conditions and has a higher computational efficiency. Furthermore, the experimental data are more suitable for actual communication scenarios; therefore, the F distribution can be used to evaluate the performance of communication systems [18]. 

The above research does not consider the optimal user jamming scheme when studying the multi-user mixed system, which has some limitations on the interference of eavesdroppers. There is also no literature on the application of F distribution to FSO links to study the physical layer security performance of mixed systems. In recent years, it is found that the simulation results of FSO link fading using F-distribution have a good fit compared with the experimental data.

To overcome the limitations of the existing techniques, this paper presents a multi-user mixed RF/FSO system with optimal user interference. The RF link experiences the Rayleigh fading distribution and the FSO link experiences the F fading distribution. SWIPT and multiple users are considered in the system. Optimal user interference, equal-gain-combining (EGC) receiving and decode-and-forward (DF) relaying schemes are invoked. In the first place, the probability density function (PDF) and cumulative distribution function (CDF) of the signal-to-interference noise ratio (SINR) of eavesdroppers under interference signals are studied, in which the interference signal is transmitted by the optimal user. Second, we obtain a unified CDF based on the system signal-to-noise ratio (SNR) of the DF trunking scheme. Based on CDF, we derive closed-form formulas for the mixed system’s SOP and ASC under the help of the Meijer G-function and the generalized Laguerre integration method, which are verified by Monte Carlo simulation. Through simulation, we analyzed the impact of factors such as the optimal number of interference-transmitting antennas, average interference SNR, energy conversion efficiency, time allocation factor, and atmospheric turbulence on the system’s safety performance.

## 2. System Model

Figure 1 shows a multi-user mixed RF/FSO system based on the optimal user interference. The system contains a multi-user source node $S_{i}$(i∈{1,…,M}), a single-input-and-single-output relay (R), a legitimate destination receiving user with a single receive antenna (D) and an illegal eavesdropper (E). The RF link experiences the Rayleigh fading distribution and the FSO link experiences the F fading distribution. The source node adopts the SWIPT technology and multi-user scheduling scheme and is equipped with an energy harvesting device. The source node utilizes the optimal user to transmit jamming signals to the illegal eavesdropper. 

Figure 2 is a schematic diagram of the time slot switching protocol for SWIPT, where ρ and T are the time switching factor and the time period of the time slot switching protocol, respectively. The relay device first transmits the RF energy signal to the source node multi-user in the first time slot ρT and the multi-user source node equipped with the energy harvesting device recognizes the received energy signal and stores it [19]. In the remaining time slot (1−ρ)T, the optimal user of the source node transmits interference signals to the eavesdropper, and the relay receives signals sent by other users of the source node with the EGC technology.

In the energy harvesting phase, the relay node R sends a radio signal $x_{RS_{i}}$ with a power of $P_{R}$, which is transmitted to each user $S_{i}$ at the transmitter through the channel of $RS_{i}$. The RF energy signal received by each user $S_{i}$ on the source node can be expressed as follows [20]:(1)$$y_{RS_{i}}=\frac{1}{\sqrt{d^{\tau }{}_{RS_{i}}}}\sqrt{P_{R}}h_{RS_{i}}x_{RS_{i}}+n_{S_{i}}$$
where $P_{R}$ and $x_{RS_{i}}$ are the transmitted power and normalized signal of the relay node, respectively, $h_{RS_{i}}$ is the channel coefficient between the users of the relay R and the source node, $d_{RS_{i}}$ represents the distance between R and $S_{i}$, τ is the path loss index, and $n_{S_{i}}$ represents the additive white Gaussian noise with zero mean and variance $\sigma_{RS_{i}}$ at each user. According to (1), it can be gathered from [21] that when noise influence is ignored, the energy stored by each user can be expressed as
(2)$$E_{S_{i}}=\rho \eta {\left|\frac{1}{\sqrt{d^{\tau }{}_{RS_{i}}}}\sqrt{P_{R}}h_{RS_{i}}\right|}^{2}T$$
where η is the energy conversion efficiency of the source node corresponding to the conversion of the RF signal into direct current (DC).

Defining the optimal user as $S_{J}$, the RF signal received by it from the relay can be expressed as
(3)$$y_{RS_{J}}=\frac{1}{\sqrt{d^{\tau }{}_{RS_{J}}}}\sqrt{P_{R}}h_{RS_{J}}x_{RS_{J}}+n_{S_{J}}$$
where $h_{RS_{J}}$ and $d_{RS_{J}}$ denote the channel coefficient between the relay R and the optimal user $S_{J}^{}$, respectively, and $n_{S_{J}}$ denotes the additive white Gaussian noise with zero mean and variance $\sigma_{RS_{J}}$ at the optimal user. 

The energy received by the optimal user $S_{J}$ is expressed as [22]
(4)$$E_{S_{J}}=\rho \eta {\left|\frac{1}{\sqrt{d^{\tau }{}_{RS_{J}}}}\sqrt{P_{R}}h_{RS_{J}}\right|}^{2}T$$

The signal received by the relay after EGC is expressed as follows:(5)$$y_{S_{G}R}=\frac{1}{2}\frac{{\left(\sum_{i=1;i\neq J}^{M}y_{S_{i}}\right)}^{2}}{\sum_{i=1;i\neq J}^{M}N_{i}}$$
where $y_{S_{i}}$ and $N_{i}$ denote the SNR and additive white Gaussian noise at each branch, respectively.

As the optimal user at the source node has multiple RF transmit antennas, it generates a matrix $W_{J}$ with an orthogonal basis of null space $N_{S_{J}}(N_{S_{J}}-1)$ using the artificial interference generation method in [23]. The ε vector has $\left(N_{S_{J}}-1\right)$ independent and identically distributed complex Gaussian random elements with a normalized variance. Subsequently, the optimal user of the source node sends $W_{\varepsilon }$ as the interference signal. When the number of the optimal user’s interference transmit antennas is greater than 1, interference signals that are transmitted into the null space will not affect the main signal, as the null space is defined as the orthogonal complement of the signal space. Under the effect of the jamming signal transmitted by the optimal user, the received signal at the illegal eavesdropper can be expressed as
(6)$$y_{S_{G}E}=\frac{\sqrt{P_{S_{G}}}}{\sqrt{d^{\tau }{}_{S_{G}E}}}h_{S_{G}E}x_{S_{G}R}+\frac{\sqrt{P_{S_{J}}}}{\sqrt{d^{\tau }{}_{S_{J}E}}}h_{S_{J}E}\frac{W_{J}}{\sqrt{N_{S_{J}}-1}}+n_{E}$$
where $h_{S_{G}E}$ is the channel coefficient between the source $S_{G}$ and the eavesdropper E, and $n_{E}$ expresses the additive white Gaussian noise with zero mean and variance $\sigma {\;}_{E}^{2}$ at the illegal eavesdropper node.

### 2.1. RF Channel Model

The jamming signal is designed in such a way that it will only affect the illegal eavesdropper without degrading the quality of the main transmission. When communicating, the signal interference to noise ratio (SINR) of the illegal eavesdropper can be written as follows:(7)$$\gamma_{S_{G}S_{J}E}=\frac{\gamma_{S_{G}E}}{\gamma_{S_{J}E}+1}$$
where $\gamma_{S_{G}E}=\frac{P_{S_{G}}}{\sigma_{E}{}^{2}d_{S_{G}E}^{\tau }}{\left|h_{S_{G}E}\right|}^{2}$, $\gamma_{S_{J}E}=\frac{P_{S_{J}}}{\sigma_{E}{}^{2}d_{S_{J}E}^{\tau }(N_{S_{J}}-1)}{\left\|h_{S_{J}E}W\right\|}^{2}$ = $\frac{\rho \eta P_{R}{\left|h_{RS_{J}}\right|}^{2}{\left\|h_{S_{J}E}W\right\|}^{2}}{(1-\rho )N_{0}(N_{S_{J}}-1)d_{RS_{J}}^{\tau }d_{S_{J}E}^{\tau }}$.

The RF links between source and relay obey the Rayleigh fading. The Instantaneous SNR $\gamma_{k}$’s PDF and CDF at the RF link can be mathematically expressed as the following formula:(8)$$f_{\gamma_{k}}(\gamma )=\frac{1}{\Omega_{k}}e\mathrm{xp}\left(\frac{-\gamma_{k}}{\Omega_{k}}\right)$$
(9)$$F_{\gamma_{k}}(\gamma )=1-e\mathrm{xp}\left(\frac{-\gamma_{k}}{\Omega_{k}}\right)$$
where $k\in \{S_{G}R,S_{G}E\}$, and $\Omega_{S_{G}R}=\frac{P_{S_{G}}\lambda_{S_{G}R}}{\sigma_{R}^{2}d_{S_{G}R}^{\tau }}$ and $\Omega_{S_{G}E}=\frac{P_{S_{G}}\lambda_{S_{G}E}}{\sigma_{E}^{2}d_{S_{G}E}^{\tau }}$ represent the average power channel gain between the corresponding RF channels, respectively. 

It is known from [24] that the CDF of the optimal user is
(10)$$\begin{array}{l} \mathrm{pr}\{\max(x_{1},x_{2},x_{3},\ldots x_{M})<x\}\\ =pr\{x_{1}<x\}pr\{x_{2}<x\}pr\{x_{3}<x\}\ldots pr\{x_{M}<x\}\\ =F^{M}(x) \end{array}$$

Using (9) and (10), the CDF of the instantaneous SNR of the optimal user interference $\gamma_{S_{J}E}$ is obtained as follows:(11)$$F_{\gamma_{S_{J}E}}(\gamma )={\left[1-\exp\left(\frac{-\gamma_{S_{J}E}}{\Omega_{S_{J}E}}\right)\right]}^{M}=\sum_{h=0}^{M}\left(\begin{array}{l} M\\ h \end{array}\right){\left[1-\exp\left(\frac{-\gamma_{S_{J}E}}{\Omega_{S_{J}E}}\right)\right]}^{h}$$

Calculating the corresponding derivative of (11) gives the PDF of the instantaneous interference SNR of the optimal user as
(12)$$f_{\gamma_{S_{J}E}}(\gamma )=M{\left[1-e\mathrm{xp}\left(\frac{-\gamma_{S_{J}E}}{\Omega_{S_{J}E}}\right)\right]}^{M-1}\frac{1}{\Omega_{S_{J}E}}\exp\left(\frac{-\gamma_{S_{J}E}}{\Omega_{S_{J}E}}\right)$$
where $\Omega_{S_{J}E}=\Psi \frac{\lambda_{S_{J}E}}{d_{S_{J}E}^{\tau }}$ is the average power channel gain between the optimal user $S_{J}^{}$ and the illegal eavesdropper E, and $\Psi =\frac{\rho \eta P_{R}N_{S_{J}}}{(1-\rho )\sigma_{E}^{2}d_{RS_{J}}^{\tau }}$.

The CDF of the instantaneous SNR $\gamma_{S_{G}S_{J}E}$ can be obtained using (7)–(9) and (12) as follows [25]:(13)$$\begin{array}{l} F_{\gamma_{S_{G}S_{J}E}}\left(\frac{\gamma_{S_{G}E}}{\gamma_{S_{J}E}+1}<\gamma_{\mathrm{th}}\right)=\int_{0}^{\infty }F_{\gamma_{S_{G}E}}\left[\gamma_{th}(\gamma_{I}+1)\right]f_{\gamma_{S_{J}E}}(\gamma_{I})d\gamma_{I}\\ =\frac{M}{\Omega_{S_{J}E}}\sum_{h=0}^{M-1}C_{M-1}^{h}{\left(-1\right)}^{h}\left[\frac{\Omega_{S_{J}E}}{h+1}-e\mathrm{xp}\left(\frac{-\gamma_{th}}{\Omega_{S_{G}E}}\right)\frac{1}{\frac{\gamma_{th}}{\Omega_{S_{G}E}}+\frac{h+1}{\Omega_{S_{J}E}}}\right] \end{array}$$

Applying the derivative operation to the CDF represented by (13), the corresponding PDF is obtained as
(14)$$f_{S_{G}S_{J}E}(\gamma )=\frac{M}{\Omega_{S_{J}E}}\sum_{h=0}^{M-1}C_{M-1}^{h}{\left(-1\right)}^{h}\left[\frac{1}{\Omega_{S_{G}E}}e\mathrm{xp}\left(\frac{-\gamma_{}}{\Omega_{S_{G}E}}\right)\left(\frac{1}{\frac{\gamma_{}}{\Omega_{S_{G}E}}+\frac{h+1}{\Omega_{S_{J}E}}}+\frac{1}{{\left(\frac{\gamma_{}}{\Omega_{S_{G}E}}+\frac{h+1}{\Omega_{S_{J}E}}\right)}^{2}}\right)\right]$$

### 2.2. FSO Channel Model

The optical signal received by the target node from the relay through the FSO link can be represented as [26,27]:(15)$$y_{D}=I_{FSO}^{\frac{r}{2}}G[1+\eta_{e}^{\frac{r}{2}}(h_{S_{G}R}x_{S_{G}R}+n_{S_{G}R})]+n_{RD}$$
where $\eta_{e}$ represents the electro-optical conversion efficiency, and G represents the gain factor. The value of r represents the type of detection method (r = 1: heterodyne detection (HD); r = 2: direct detection (DD)), $y_{S_{G}R}$ indicating the signal received by the source node. $I_{FSO}$ represents the channel coefficient of the FSO link and $n_{RD}$ represents the zero-mean AWGN with variance $\sigma_{RD}^{2}$.

The PDF and CDF of the FSO fading channel are expressed as follows [28]: (16)$$f_{\gamma_{RD}}(\gamma_{RD})=\frac{\xi_{\mathrm{mod}}^{2}}{r\Gamma (a)\Gamma (b)\gamma_{RD}}G_{2,2}^{2,1}\left[\frac{a\phi \gamma_{RD}^{\frac{1}{r}}}{\bar{(b-1)\mu_{r}^{\frac{1}{r}}}}|\begin{array}{l} 1-b,\xi_{\mathrm{mod}}^{2}+1\\ a,\xi_{\mathrm{mod}}^{2} \end{array}\right]$$
(17)$$F_{\gamma_{RD}}(\gamma_{RD})=1-\frac{\xi_{\mathrm{mod}}^{2}}{r\Gamma (a)\Gamma (b)}H_{3,3}^{3,1}\left[\frac{a\phi \gamma_{RD}^{\frac{1}{r}}}{\bar{(b-1)\mu_{r}^{\frac{1}{r}}}}|\begin{array}{l} (1-b,1),(\xi_{\mathrm{mod}}^{2}+1,1),\left(1,\frac{1}{r}\right)\\ \left(0,\frac{1}{r}\right),(a,1),(\xi_{\mathrm{mod}}^{2},1) \end{array}\right]$$
where $\mu_{r}$ is the average telecom noise ratio, r = 1 represents the heterodyne detection (HD), and r = 2 represents the direct detection (DD).
$$\mu_{1}=\mu_{HD}=\bar{\gamma_{RD}}(HD),\;\mu_{2}=\mu_{DD}=\frac{a\phi \left(\xi_{\mathrm{mod}}^{2}+2\right)\left(b-2\right)\bar{\gamma_{RD}}(DD)}{\left[(b-1)(a+1)\left(\xi_{\mathrm{mod}}^{2}+1\right)\right]},\;\phi =\frac{\xi_{\mathrm{mod}}^{2}}{\xi_{\mathrm{mod}}^{2}+1}$$

In the above expressions, $\xi_{\mathrm{mod}}^{2}$ is related to the equivalent beam width $w_{Leq}$, and a and b denote the turbulence parameters. The value of $\xi_{\mathrm{mod}}^{2}$ increases as the pointing error decreases. When there is no pointing error, $\xi_{\mathrm{mod}}^{2}\to \infty$. 

## 3. End-to-End SNR Statistics

Under DF relaying, the end-to-end instantaneous SNR is denoted by
(18)$$\gamma_{S_{G}RD}=\frac{\gamma_{S_{G}R}\gamma_{RD}}{\gamma_{S_{G}R}+\gamma_{RD}+1}≅\min(\gamma_{S_{G}R},\gamma_{RD})$$

The mixed channel’s CDF is denoted by [29]
(19)$$F_{\gamma_{S_{G}RD}}=P_{r}\left[\min(\gamma_{S_{G}R},\gamma_{RD})<\gamma \right]=F_{\gamma_{S_{G}R}}(\gamma )+F_{\gamma_{RD}}(\gamma )-F_{\gamma_{S_{G}R}}(\gamma )F_{\gamma_{RD}}(\gamma )$$

Substituting (9) and (17) into (19) results in the following equation
(20)$$F_{\gamma_{S_{G}RD}}(\gamma )=1-\exp\left(\frac{-\gamma }{\Omega_{S_{G}R}}\right)\frac{\xi_{\mathrm{mod}}^{2}}{r\Gamma (a)\Gamma (b)}H_{3,3}^{3,1}\left[\frac{a\phi \gamma_{}^{\frac{1}{r}}}{(b-1)\mu_{r}^{\frac{1}{r}}}|\begin{array}{l} (1-b,1),\left(\xi_{\mathrm{mod}}^{2}+1,1\right),\left(1,\frac{1}{r}\right)\\ \left(0,\frac{1}{r}\right),(a,1),\left(\xi_{\mathrm{mod}}^{2},1\right) \end{array}\right]$$

## 4. Secrecy Outage Probability Analysis

The Secrecy Outage Probability (SOP) is a benchmark for secrecy, indicating the probability of the occurrence of an event where the instantaneous secrecy capacity falls below the target secrecy rate $R_{S}$. The mixed system’s SOP lower bound formula is defined as follows [30]:(21)$$P_{\mathrm{out}}^{L}(R_{S})=\int_{0}^{\infty }F_{S_{G}RD}(\theta \gamma )f_{S_{G}S_{J}E}(\gamma )d\gamma$$
where $\theta =\exp(R_{s})$. Substituting (8) and (20) into (21), and then combining the relevant expressions given in [31] and the generalized Laguerre polynomial [32], after some simplifying mathematical operations, the SOP can be obtained as follows:(22)$$\begin{array}{l} P_{out}^{L}(R_{S})=\sum_{j=1}^{n}H_{j}Y_{j}^{0.5}\exp\left(Y_{j}\right)\frac{M}{\Omega_{S_{J}E}}\sum_{h=0}^{M-1}C_{M-1}^{h}{\left(-1\right)}^{h}\\ \times \left[\frac{1}{\Omega_{S_{G}E}}\exp\left(\frac{-Y_{j}}{\Omega_{S_{G}E}}\right)\left(\frac{1}{\frac{Y_{j}}{\Omega_{S_{G}E}}+\frac{h+1}{\Omega_{S_{J}E}}}+\frac{1}{{\left(\frac{Y_{j}}{\Omega_{S_{G}E}}+\frac{h+1}{\Omega_{S_{J}E}}\right)}^{2}}\right)\right]\\ \times \left\{1-\exp\left(\frac{-\theta Y_{j}}{\Omega_{S_{G}R}}\right)\frac{\xi_{\mathrm{mod}}^{2}}{r\Gamma (a)\Gamma (b)}H_{3,3}^{3,1}\left[\frac{a\phi \theta Y_{j}{}^{\frac{1}{r}}}{(b-1)\mu_{r}^{\frac{1}{r}}}|\begin{array}{l} (1-b,1),\left(\xi_{\mathrm{mod}}^{2}+1,1\right),\left(1,\frac{1}{r}\right)\\ \left(0,\frac{1}{r}\right),(a,1),\left(\xi_{\mathrm{mod}}^{2},1\right) \end{array}\right]\right\} \end{array}$$

$H_{j}=\frac{\Gamma (n+1/2)Y_{j}}{n!{\left(n+1\right)}^{2}{\left[L_{n}^{\left(-1/2\right)}(Y_{j})\right]}^{2}}$, $Y_{j}$ means the jth root of the generalized Laguerre polynomial $L_{n}^{\left(-\frac{1}{2}\right)}(Y)$.

## 5. Average Secrecy Capacity Analysis

The Average Secrecy Capacity (ASC) is a critical metric for assessing the active-eavesdropping’s security performance and can be expressed as follows:(23)$$\bar{C_{S}}=\int_{0}^{\infty }\frac{F_{S_{G}S_{J}E}(\gamma )}{1+\gamma }\left[1-F_{S_{G}RD}(\gamma )\right]d\gamma$$

Substituting (13) and (20) into (23) and after a few mathematical simplifications described above, we obtain
(24)$$\begin{array}{l} \bar{C_{S}}=\sum_{j=1}^{n}H_{j}Y_{j}^{0.5}\exp\left(Y_{j}\right)\frac{1}{1+Y_{j}}\frac{M}{\Omega_{S_{J}E}}\sum_{h=0}^{M-1}C_{M-1}^{h}{\left(-1\right)}^{h}\\ \times \left(\frac{\Omega_{S_{J}E}}{h+1}-\exp\left(\frac{-Y_{j}}{\Omega_{S_{G}E}}\right)\frac{1}{\frac{Y_{j}}{\Omega_{S_{G}E}}+\frac{h+1}{\Omega_{S_{J}E}}}\right)\exp\left(\frac{-Y_{j}}{\Omega_{S_{G}R}}\right)\frac{\xi_{\mathrm{mod}}^{2}}{r\Gamma (a)\Gamma (b)}H_{3,3}^{3,1}\left[\frac{a\phi Y_{j}^{\frac{1}{r}}}{(b-1)\mu_{r}^{\frac{1}{r}}}|\begin{array}{l} (1-b,1),\left(\xi_{\mathrm{mod}}^{2}+1,1\right),\left(1,\frac{1}{r}\right)\\ \left(1,\frac{1}{r}\right),(a,1),\left(\xi_{\mathrm{mod}}^{2},1\right) \end{array}\right] \end{array}$$

## 6. Results and Analysis

In this part, the simulation data of the multi-user mixed RF/FSO system under diverse parameters with optimal user interference are presented. Monte Carlo simulations verify the numerical results’ accuracy. The FSO link distance is 100 m and the wavelength is 785 nm, the optical wave number k=2π/λ; $C_{n}^{2}=10^{-11}\;m^{-2/3}$ is the atmospheric refractive index structure constant of medium turbulence, and the target secrecy rate $R_{S}=0.01\mathrm{nat}/s$. Furthermore, the Instantaneous SNR of the FSO link is $\gamma_{RD}=20\;\mathrm{dB}$, the instantaneous SNR of the RF link is $\gamma_{SR}=15\;\mathrm{dB}$, and the instantaneous SNR of the eavesdropping link is $\gamma_{SE}=-10\;\mathrm{dB}$. On the RF link, $d_{SE}=d_{SR}=10\;m$, and $d_{RE}=5\;m$. The other parameters are η=0.8, ρ=0.8, $N_{J}=2$, r=1, a=1.83, b=3.94, and $\xi_{\mathrm{mod}}=0.84$. The specific simulation parameters are shown in Table 1.

The above parameter values are used in the following simulations. When the generalized Laguerre orthogonal numerical integration method is used, j is considered as 30 in order to make the series converge. The Monte Carlo simulation results confirm the validity of the analytical expressions. The numerical and simulation results are in good agreement, which verifies the accuracy of the proposed expressions.

Figure 3 shows the mixed system’s SOP as a function of the variation of the RF link’s Instantaneous SNR $\lambda_{SR}$ for diverse numbers of the optimal user’s interfering antennas. It can be known that the mixed system’s SOP with different number of interfering antennas $N_{J}$ decreases as the $\lambda_{SR}$ increases. When $\lambda_{SR}=30\;\mathrm{dB}$ and $N_{J}=2,4,6$ and 8, the mixed system’s SOP are $9.7\times 10^{-4}$, $3.0\times 10^{-4}$, $1.3\times 10^{-4}$ and $6.9\times 10^{-5}$, respectively. This set of simulation data shows the mixed system’s SOP decreases as the number of interfering antennas of the optimal user increases. Therefore, it can be gathered that the mixed system’s security performance could be effectively enhanced by enlarging the optimal user interference antennas’ number at the source node.

As the spatial degree of freedom of the optimal user increases with an increase in the number of optimal user interference antennas, the eavesdropping channel quality degrades based on its state information at the eavesdropping node. For the same value of SOP, increasing the optimal user’s number of jamming antennas results in lower transmission power for each antenna in the scheme. In addition, the cost of increasing the antennas’ number is lower than that of adding jammers outside the mixed RF/FSO system. Increasing the number of jamming antennas provides a good schedule for the design of the optimal user transmitting jamming signals and reducing the power consumption of the source node.

The relationship between the SOP and the RF link’s instantaneous SNR $\lambda_{SR}$ in a mixed RF/FSO system is shown in Figure 4. The system is based on multi-user SWIPT, and the illegal eavesdropper experiences interference from the optimal user. Based on the simulation results, increasing $\lambda_{SR}$ reduces the SOP of the system in a gradual manner. When $\lambda_{SR}=30\;\mathrm{dB}$ and $\lambda_{JE}=$2, 4, 6 and 8 dB, the mixed system’s SOP are $9.6\times 10^{-4}$, $2.9\times 10^{-4}$, $1.4\times 10^{-4}$ and $4.0\times 10^{-5}$. Based on the findings, increasing $\lambda_{JE}$ decreases the system’s SOP, indicating that the optimal user’s jamming signal to the eavesdropper improves PLS performance in the mixed system. Increasing the interference SNR of the optimal user can reduce eavesdropping channel quality, but it is restricted by transmission power. Thus, introducing optimal user interference and selecting appropriate transmit power can enhance the mixed system’s secrecy performance.

Figure 5 depicts the mixed system’s SOP as a function of the RF link’s Instantaneous SNR $\lambda_{SR}$ when the optimal user at the source node adopts different values of energy conversion efficiency η. The figure shows that the mixed system’s SOP diminishes as $\lambda_{SR}$ increases for different values of η. When $\lambda_{SR}=30\;\mathrm{dB}$ and η= 0.3, 0.5, 0.7 and 0.9, the mixed system’s SOP are $9.7\times 10^{-4}$, $4.2\times 10^{-4}$, $2.2\times 10^{-4}$ and $1.3\times 10^{-4}$, respectively. As the energy conversion efficiency of the optimal user increases, the SOP of the system decreases.

The energy harvesting multi-user source node efficiently converts the relay’s RF energy signal into DC, boosting both the stored energy and interference power of the optimal user relative to the illegal eavesdropper. Consequently, the jamming effect is improved, the communication quality of the illegal eavesdropper deteriorates, and the mixed system’s PLS performance is improved. By improving the optimal user energy conversion efficiency scheme, the collected energy is more stable and that makes the system more secure.

Figure 6 shows the ASC of the mixed RF/FSO system as a function of the instantaneous SNR $\lambda_{SR}$ of the RF link when the optimal user adopts different time allocation factors ρ. The simulation results show that the ASC of the system increases as $\lambda_{SR}$ increases. When $\lambda_{SR}=30\;\mathrm{dB}$ and ρ=0.1, 0.3, 0.5 and 0.7, the ASC values of the system are 1.8957, 1.9088, 1.9242 and 1.9464, respectively. This set of simulation data shows that the ASC of the mixed RF/FSO system can be effectively increased by increasing the time allocation factor ρ. This is because as the time allocation factor ρ increases, the time taken by the relay to transmit the RF energy signal to the optimal user of the source node becomes longer in the first time slot and the RF energy received by the source node increases. The energy stored is higher, and the power of the optimal user to transmit interference signal to the eavesdropper is enhanced at the same time, which increases the interference effect of the optimal user on the illegal eavesdropper. Therefore, the communication quality and the mixed system’s PLS performance are improved.

In Figure 7, the ASC is plotted against the RF link’s instantaneous SNR $\lambda_{SR}$ in the mixed system, while the eavesdropper experiences varying levels of interference SNR $\lambda_{JE}$ from the optimal user. Simulation results show an ascending trend of the ASC alongside increasing $\lambda_{JE}$. When $\lambda_{SR}=30\;\mathrm{dB}$, $\lambda_{JE}=$2, 3, 6 and 9 dB, the system’s ASC values of 1.95, 1.96, 1.98, and 2.01 support the conclusion that higher interference SNR $\lambda_{JE}$ from the optimal user boosts the ASC to a certain degree. This affirms that transmitting jamming signals from the optimal user to the eavesdropper enhances the security of the communication system, and higher transmission power bolsters the PLS performance.

The graphic in Figure 8 plots the mixed system’s ASC against the FSO link’s instantaneous SNR $\lambda_{RD}$ model which is based on the F distribution under the optimal user interference for different turbulence conditions. The figure shows that when $\lambda_{RD}$ = 30 dB, the ASC of the system is 2.08 in weak turbulence conditions of a = 2.33 and b = 4.53. When a = 1.83 and b = 3.94, the ASC of the system is 2.02. The ASC of the system is 1.94 in the case of strong turbulence with a = 1.43 and b = 3.53. As the atmospheric turbulence intensity increases, the ASC of the system decreases. This shows that the FSO link is sensitive to weather conditions, and a bad weather environment will seriously affect the security performance of the FSO link.

## 7. Conclusions

In this paper, the PLS performance of a mixed RF/FSO communication system with multi-user SWIPT under optimal user interference was studied. The mixed system’s SOP and ASC were researched theoretically and through simulations, with the derived expressions’ validity verified using the Monte Carlo method. The impacts of the number of jamming antennas, average jamming SNR, energy conversion efficiency, time allocation factor and atmospheric turbulence on the mixed system’s security performance were mainly studied. Simulation results indicated that as the RF link’s instantaneous SNR increased, the system’s SOP gradually decreased while the ASC increased. When the number of antennas from which the optimal user transmitted the interference signals was enlarged, the mixed system’s SOP was reduced. Increasing the number of the optimal user’s jamming antennas could enhance the interference effect and reduce the transmit power of each single antenna, which was an effective scheme to enhance the mixed system’s security performance. When the optimal user interference SNR increased, the mixed system’s SOP decreased and ASC gradually increased. This showed that the interference effect on the eavesdropper could be enhanced by increasing the optimal user interference SNR, which improved the mixed system’s security performance. On the other hand, improving the energy conversion efficiency of the optimal user simultaneously enabled the energy harvesting device to store more energy and increase the power of the interference signal transmitted by the energy harvesting device, which in turn enhanced the secrecy ability of the mixed system. By using the SWIPT technology to adjust the time allocation factor, the optimal user energy harvester could prolong the receiving time of the RF energy signal sent by the relay. This phenomenon increased the energy required by the optimal user to transmit the interference signals to the illegal eavesdropper and enhance the mixed system’s security performance. Additionally, the system’s ASC demonstrates an increase as the FSO link’s instantaneous SNR increases with optimal user interference. Notably, when turbulence factors a, b of the F distribution are adjusted from strong to weak turbulence, the system’s ASC increases, suggesting that improved weather conditions can enhance the PLS performance. It can be seen that the mixed RF/FSO system based on optimal user interference limits the eavesdropping SNR of potential illegal eavesdrops to a large extent, thereby increasing the average security capacity of the mixed system and reducing its security outage probability. This scheme not only improves the communication quality of the mixed system, but also significantly improves the physical layer security of the system, which is of great significance to solve the problem of interception in the mixed system. With the development of science and technology, there may be potential eavesdroppers in the FSO link of mixed system. Therefore, the application of the optimal user jamming scheme into the mixed system with eavesdroppers in both links in the future has important research value to improve the security performance of the system.

## Figures and Tables

**Figure 1 sensors-23-06523-f001:**
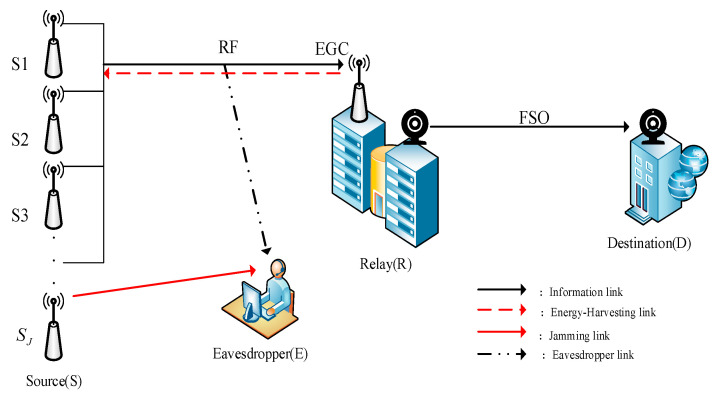
A multi-user mixed RF/FSO system based on optimal user interference.

**Figure 2 sensors-23-06523-f002:**
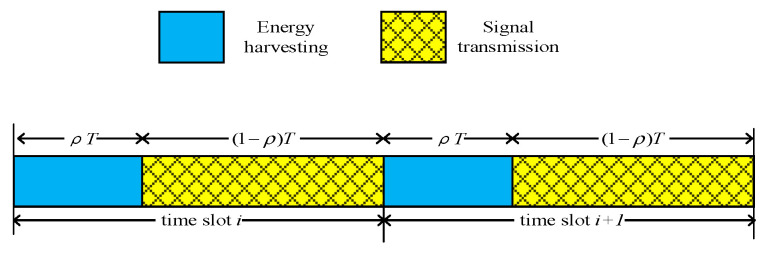
Time slot switching (TS) protocol for SWIPT.

**Figure 3 sensors-23-06523-f003:**
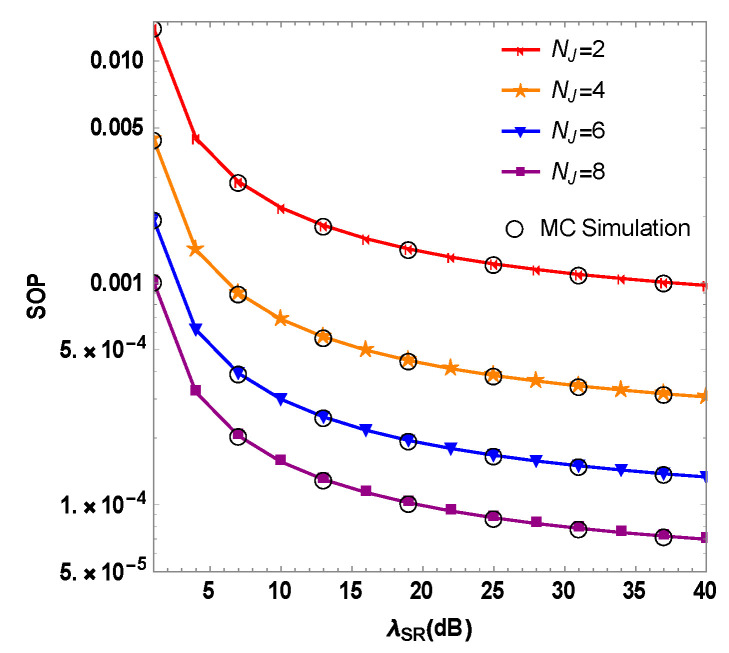
Simulation of the SOP and the Instantaneous SNR of RF link for the RF/FSO system with the optimal user’s different numbers of interference antennas $N_{J}$.

**Figure 4 sensors-23-06523-f004:**
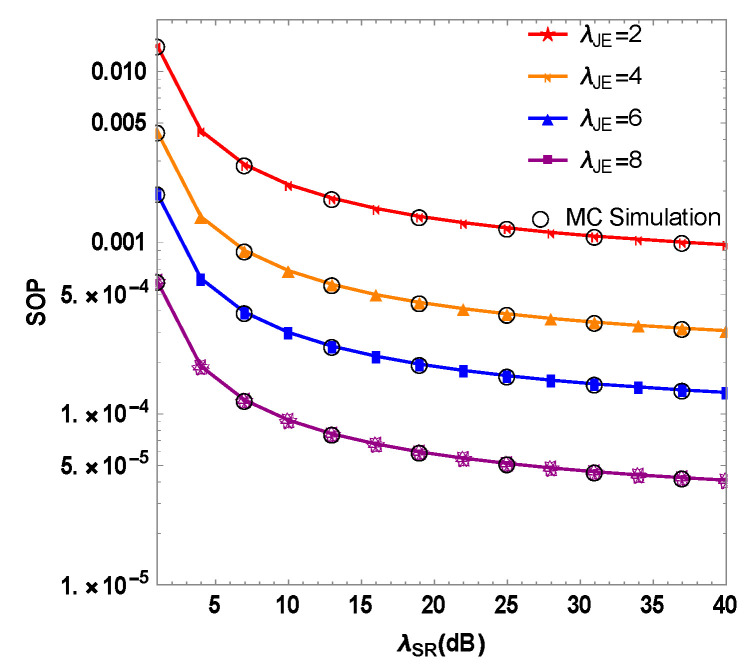
Simulation of the SOP and the instantaneous SNR of the RF link of the RF/FSO system under different interference SNR $\lambda_{JE}$ of the optimal user.

**Figure 5 sensors-23-06523-f005:**
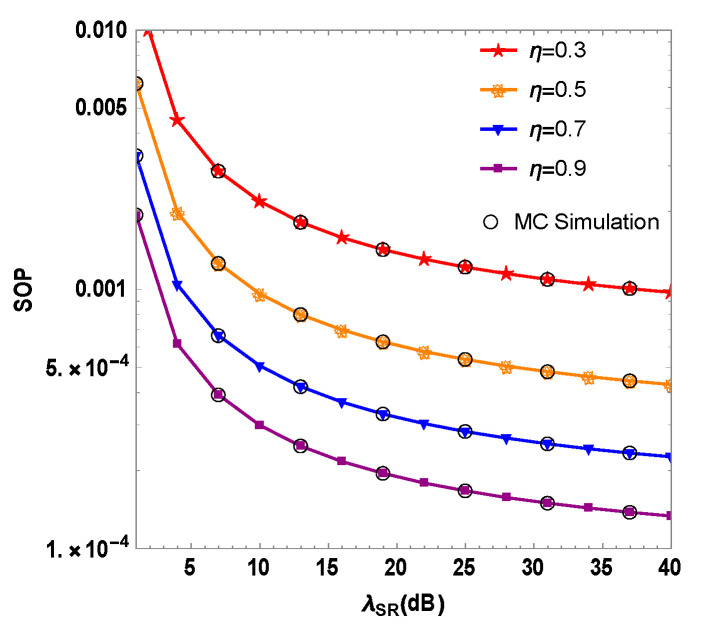
Simulation of the security outage probability and the instantaneous signal-to-noise ratio of the RF link in RF/FSO system under different energy conversion efficiency η of the optimal user.

**Figure 6 sensors-23-06523-f006:**
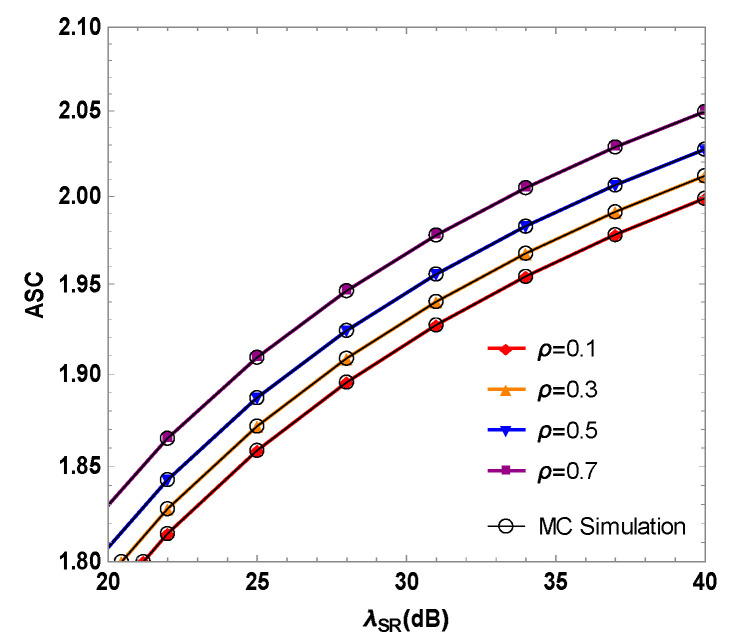
Simulation of the average secrecy capacity and the instantaneous SNR of RF link in RF/FSO system when the optimal user uses different time allocation factors ρ.

**Figure 7 sensors-23-06523-f007:**
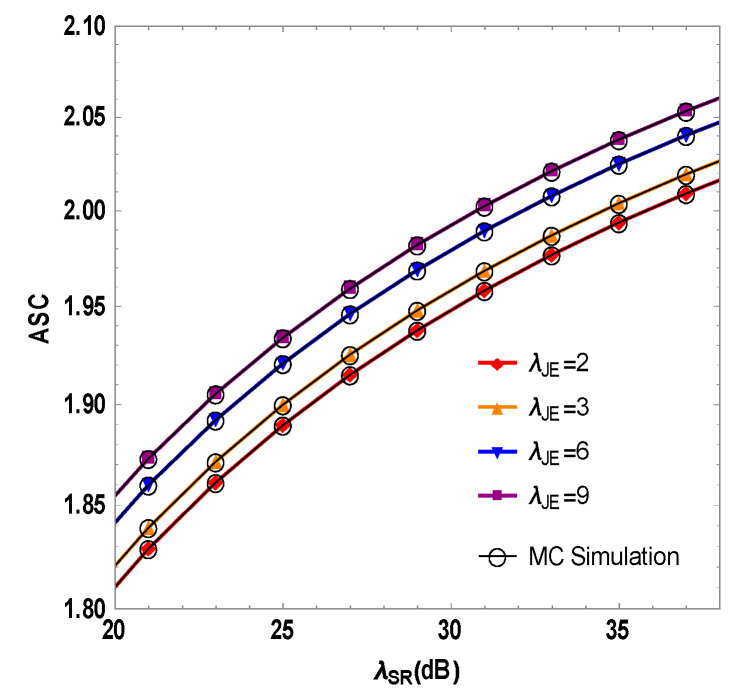
Simulation of the average secrecy capacity and the instantaneous SNR of the RF link of the RF/FSO system under different interference SNRS $\lambda_{JE}$ of the optimal user.

**Figure 8 sensors-23-06523-f008:**
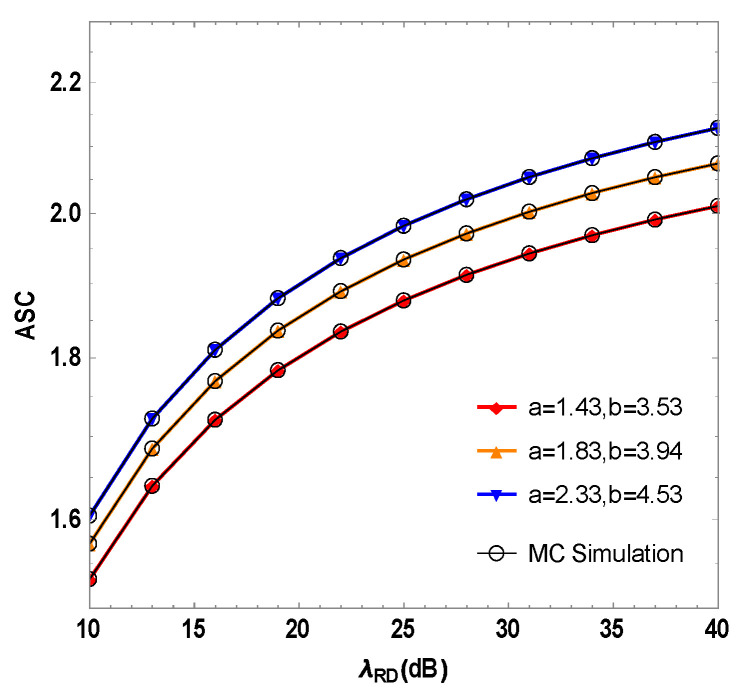
Simulation of the ASC of the RF/FSO system with different turbulence factors a, b and the instantaneous SNR $\lambda_{RD}$ of the FSO link under optimal user interference.

**Table 1 sensors-23-06523-t001:** Selection of simulation parameters for the mixed RF/FSO system.

System Parameters	Symbol	Value	System Parameters	Symbol	Value
Laser wavelength	λ	785 nm	Link distance per hop	*d*	100 m
Weak turbulence positive parameter	a	2.33	Weak turbulence fading parameters	b	4.53
Medium turbulence positive parameter	a	1.83	Medium turbulence fading parameters	b	3.94
Strong turbulence positive parameter	a	1.43	Strong turbulence fading parameters	b	3.53
Electro-optic conversion efficiency	ω	0.8	target secrecy rate	$R_{S}$	0.01 nat/s
atmospheric refractive index structure constant	$C_{n}^{2}$	$10^{-11}\;m^{-2/3}$	Energy conversion efficiency	η	0.8
Time switching factor	ρ	0.8	the instantaneous SNR of the FSO link	$\gamma_{RD}$	20 dB
the instantaneous SNR of the RF link	$\gamma_{SR}$	15 dB	Number of interfering antennas	$N_{J}$	2

## Data Availability

Not applicable.

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
