# Peer review of "Physical Layer Security Performance of Multi-User Mixed Radio-Frequency/Free-Space-Optics System Based on Optimal User Interference"

_sensors, 2023, doi:10.3390/s23146523_

Round 1

Reviewer 1 Report

The comments on the manuscript entitled "On PLS Performance of multi-user mixed RF/FSO system based on optimal user interference", by Shen et al.:

(1) All abbreviations should be defined. Please revise the title for PLS. Also, it is recommended to add a list of all abbreviations at the end of manuscript.

(2) Please give proper reference(s) for Eqs. (1) and (4).

(3) Authors mentioned that the numerical and simulation results are in good agreement, which verifies the accuracy of the proposed expressions. However, these results should be verified by experimental results.

(4) The literature review is not written depth. The literature survey needs to be completed. Some important systems can be considered in Introduction section. For this reason, the following references are suggested: 10.1016/j.jnca.2021.103311, 10.1016/j.optcom.2018.08.001, 10.1155/2023/8685686.

(5) The parameters used for simulation of the secrecy outage probability (SOP) should be presented and addressed in more details.

(6) It is suggested to give a block diagram of channel parameters for readers.

(7) Finally, the manuscript needs to be polished and edited.

Author Response

Dear reviewer:

First of all, thank you very much for your valuable suggestions for the revision of the manuscript submitted by us. Thank you very much for your attention and review work. We have carefully revised the manuscript after your comments. The following is my modification instructions.

  1. We've changed the title and added a table with all the abbreviations at the end of the article.
  2. For Equations (1) and (4), we give appropriate references.
  3. At present, we are still in the theoretical research. So we used Monte Carlo simulation to verify the accuracy of the results. In the next step, we will conduct experimental simulation according to the actual situation to analyze the difference between the simulation data and the real experimental data, so as to optimize our scheme.
  4. Thank you for your suggestions. We have carefully read these literatures and decided to cite them to make the article more theoretical basis and content richer.
  5. We have modified to give detailed communication simulation parameters at the beginning of the simulation analysis.
  6. We put together a communication parameter block diagram for readers to read better.
  7. We have revised and polished the entire manuscript to improve its quality and readability.

Kind Regards!

Sincerely yours,

Yi Wang

Reviewer 2 Report

Please refer to report attached as pdf file.

The quality of the English adopted throughout the text is in general good.
However, another round of proofreading could be useful.

Author Response

Dear reviewer:

First of all, thank you very much for your valuable suggestions for the revision of the manuscript submitted by us. Thank you very much for your attention and review work. We have carefully revised the manuscript after your comments. The following is my modification instructions.

  1. To make the formulas clearer, we have re-edited the formulas appearing throughout the paper.
  2. I have given a more detailed background on the hybrid RF/FSO system involved in the introduction.
  3. At the end of the introduction, I highlight my contribution to this manuscript in a single paragraph with a detailed summary of the shortcomings of the existing research and where it can be improved.
  4. We have taken your suggestion and read and cited the references you mentioned in the study of FSO links.
  5. We deeply dissect the simulation results and theoretical basis in the last paragraph, and considering the limitations of the study, we also propose some solutions, hoping that these problems can be practically solved in future work.

Kind Regards!

Sincerely yours,

Yi Wang

Reviewer 3 Report

The paper presents the PLS Performance of multi-user mixed RF/FSO system based on optimal user interference. The paper is interesting, however some matters need to be addressed before publication.

1. A better explanation of your contribution needs to be added in the Introduction Section. Please set differences of your work with respect to previous work. Please highligh the advantages, benefits and contributions of the new tecnique and results.

2. Provide more details of the methodology used.

3. It is needed a better description of the results. Please highligh the benefits, the advantages and contributions of your proposal.

Author Response

Dear reviewer:

First of all, thank you very much for your valuable suggestions for the revision of the manuscript submitted by us. Thank you very much for your attention and review work. We have carefully revised the manuscript after your comments. The following is my modification instructions.

  1. At the end of the introduction, I highlight my contribution to this manuscript in a single paragraph with a detailed summary of the shortcomings of the existing research and where it can be improved.
  2. We have modified the methods used in detail and added references for readers to read and understand
  3. We deeply dissect the simulation results and theoretical basis in the last paragraph. We also detail the advantages of the proposed scheme in the last paragraph and considering the limitations of the study. We also propose some solutions, hoping that these problems can be practically solved in future work.

Kind Regards!

Sincerely yours,

Yi Wang

Reviewer 4 Report

it is a good paper

Author Response

Dear reviewer:

Thank you very much for your valuable suggestions for the revision of the manuscript submitted by us. Thank you very much for your attention and review work. We have carefully revised the manuscript after your comments. The following is my modification instructions.

Kind Regards!

Sincerely yours,

Yi Wang

Round 2

Reviewer 1 Report

The authors have tried to revise the manuscript according to the comments and the new version can be accepted for publication. However, as a minor revision, the position of Table 2 should be corrected (and table caption).

Reviewer 2 Report

The Authors have effectively addressed my comments and I think the overall quality and clarity of the work have been improved. Thus, I can suggest now to consider this manuscript for possible acceptance.